# Symptomatic Joint Hypermobility Is Associated with Low Back Pain: A National Adolescents Cohort Study

**DOI:** 10.3390/jcm11175105

**Published:** 2022-08-30

**Authors:** Oded Hershkovich, Barak Gordon, Estela Derazne, Dorit Tzur, Arnon Afek, Raphael Lotan

**Affiliations:** 1Department of Orthopedic Surgery, Wolfson Medical Center, Ha-Lokhamim St 62, Holon 5822012, Israel; 2Medical Corps, Israeli Defense Forces, Jerusalem 9112102, Israel; 3Sackler Faculty of Medicine, Tel Aviv University, Tel Aviv 6997801, Israel; 4Department of Military Medicine, Faculty of Medicine, Institute for Research in Military Medicine (IRMM), The Hebrew University of Jerusalem, Jerusalem 9112102, Israel

**Keywords:** low back pain, hypermobility, adolescents, joints

## Abstract

**Background**: Low back pain (LBP) is a widespread medical complaint affecting many people worldwide and costing billions. Studies suggest a link between LBP and joint hypermobility. This study aimed to examine the association between symptomatic joint hypermobility (SJH), LBP, and gender. **Methods:** Data were obtained from a medical database containing 17-year-old candidates’ records before recruitment into mandatory military service. According to the Regulations of Medical Fitness Determination, information on disability codes associated with LBP and SJH was retrieved. **Results:** According to this national survey, the prevalence of SJH is 0.11% (1355 cases out of 1,220,073 subjects). LBP was identified in 3.7% of the cohort (44,755 subjects). Subjects were further subdivided into LBP without objective findings (LBPWF) (3.5%) and LBP with objective findings (LBPOF) (0.2%). The association between SJH and LBP was examined: the Odds Ratio (OR) was 2.912 (*p* < 0.0001). The odds rations for LBPWF and LBPOF were further calculated to be 2.914 (*p* < 0.000) and 2.876 (*p* < 0.000), respectively. Subjects with SJH were almost three times more prone to LBPWF and LBPOF. **Conclusion:** SJH is strongly associated with LBP in young adults. Further pathophysiological research is needed.

## 1. Introduction

Low back pain (LBP) is one of the most frequent medical complaints in children, adolescents, and adults. The prevalence of LBP in all age groups is still controversial but relatively high [1,2]. LBP diagnosis relies mainly on patient-reported symptoms, as physical examination and imaging studies are frequently non-diagnostic. Therefore, LBP’s etiology often remains unknown [3,4].

Hypermobility of the joints might be linked to LBP pathogenesis, but evidence-based confirmation of its possible role is lacking. Hypermobility might increase the spinal motion segment’s mechanical loads through an increased range of motion by facet capsule laxity, intervertebral ligament laxity, and increased intervertebral disc motion (annular laxity) [5,6], potentially leading to earlier degenerative changes through excessive load bearing [7,8,9,10]. Another hypothesis suggested is that the biological factors associated with hypermobility are associated with processes leading to LBP [11,12,13].

The prevalence of asymptomatic joint hypermobility has been studied in various populations. Most people with joint hypermobility are asymptomatic [14,15,16], with only a small percentage having specific complaints leading to a formal diagnosis of symptomatic hypermobility of joints. This explains the literature’s wide range of reported SJH prevalences, ranging from 1% to 25.4% in males and 7% to 38.5% in asymptomatic females [17,18], depending on age, ethnicity, and diagnostic criteria used [19]. There is also uncertainty about the prevalence of joint hypermobility and its influence on health. Reports of joint hypermobility prevalence must be viewed carefully due to the variability in hypermobility diagnostic criteria used.

Joint hypermobility is more prevalent in young adults [15,20], especially in women [21]. In contrast to hyperflexibilty, hypermobility is associated with the looseness of ligaments and joint capsules rather than excessive lengthening of the musculotendinous unit. However, both may present with joint hypermobility and flexibility. In some situations, joint hypermobility may be advantageous for performance (for example, gymnasts or musicians), but individuals with joint hypermobility can experience considerable pain, disability, and decreased quality of life [22,23,24]. Disproportionate joint mobility combined with a lack of muscular control may alter soft-tissue, cartilaginous, and osseous stresses during activities [15,25]. Therefore, joint hypermobility is considered a potential risk for LBP and early spine degeneration.

Even though LBP is a common and disabling medical condition, its etiology is still not fully understood and its true prevalence throughout life, and especially during the first decades of life, is still not fully revealed. The association between LBP and SJH is still an open debate that justifies further studies into the possible pathophysiological basis [8,12]. Better discerning LBP etiology might aid in developing specific treatment strategies.

We hypothesized that SJH is associated with LBP, probably more so in females. This study examines the prevalence of LBP, with and without objective findings, and symptomatic joint hypermobility (SJH) in a national cohort of adolescents.

## 2. Materials and Methods

### 2.1. Source of Data

At 17 years of age, most Israelis, both male and female, are mandated by law to undergo a comprehensive medical evaluation at a military recruitment center for medical classification before recruitment. The evaluation process includes a medical questionnaire filled in by the candidate and a medical report signed by the candidate’s primary care physician. This questionnaire includes general questions referring to previous musculoskeletal fractures, dislocation and sprains, and rheumatic diseases. The candidates then undergo a complete anamnesis and physical examination by army physicians and are referred to medical specialists or additional imaging tests as needed. After completing the medical evaluation, each subject is assigned a number on a global medical profile scale, with numerical codes representing the subject’s diagnosis and medical status. These codes are defined by the Israeli Defense Forces Regulations for Medical Fitness Determination and describe various medical conditions.

The study’s data were extracted from the Israeli Medical Corps database, as approved by the Israel Defense Forces Medical Corps Institutional Review Board. Non-identified recruit medical data were collected. No informed consent was required for this epidemiological study.

### 2.2. Study Population

This is a retrospective epidemiological study with complete medical data including 1,220,073 adolescents evaluated by regional army recruitment centers since 1998.

### 2.3. Cohort Assignment

Subjects diagnosed with LBP were classified into two groups. The first group was LBP without corroborative objective findings (LBFWF) (e.g., neurological deficit or radicular irritation symptoms) on physical examination or imaging studies (i.e., computerized tomography, magnetic resonance imaging, or myelography). In contrast, the second group was comprised of recruits that suffered LBP with objective findings (LBPOF) correlating to the patient’s diagnosis (e.g., herniated disk or spinal stenosis).

A two-step process diagnosed SJH; the first was the medical screening questionnaire completed by the adolescent stating hypermobility with a documented history of relevant clinical features (joint pain, back pain, neck pain, recurrent ankle sprains, etc.). The second step was hypermobility confirmation by Beighton’s score of 5/9 or higher [17,26,27] on physical examination by a rheumatologist. Connective tissue diseases such as Marfan’s disease [28] or Ehlers–Danlos syndrome [29] were excluded from this study. Undiagnosed asymptomatic hypermobile subjects not classified by a disability code inherently were not included in this study.

### 2.4. Data Analysis

SJH and LBP associations were assessed by logistic regression analyses that applied the following models: binary models when LBP was considered as a dichotomous variable and multinomial model analysis with no LBP as the base category for comparison when LBP was classified as LBPWF or LBPOF. Logistic regression analysis results were presented as odds ratios, 95% confidence intervals, and *p* values. Statistical analyses were performed using SPSS software, version 19.0 (SPSS Inc., Chicago, IL, USA).

## 3. Results

The study included 1,220,073 adolescents, comprising 57.15% males (697,272) and 42.85% females (522,801). Most of the cohort were healthy young individuals without SJH or LBP. The cohort’s LBP prevalence was 3.7% (44,755 adolescents), while 0.11% (1355 adolescents) suffered from SJH. When categorizing LBP, 42,406 (3.5%) subjects had LBPWF and 2349 (0.2%) suffered LBPOF, with correlating anatomical findings. Of the 1355 subjects diagnosed with SJH, 10% had concomitant LBP, compared to only 3.7% in the non-SJH cohort (*p* < 0.0001). Most (1220) subjects had LBP without objective findings. LBPOF was rare in SJH and non-SJH groups of adolescents (0.2%). LBPOF was 2.5 times more prevalent in the SJH group of patients (0.5%, *p* < 0.0001) (Table 1). Gender was not associated with symptomatic joint hypermobility (OR = 1.098, CI 0.989–1.223; *p* = 0.09).

When examining the association between SJH and LBP with and without objective findings, we found a strong association between the groups, with an Odds Ratio for LBP and SJH of 2.912 (CI 2.437–3.479; *p* < 0.0001). Further analysis according to LBP versus LBPWF and LBPOF resulted in odds ratios of 2.914 (CI 2.428–3.497; *p* < 0.000) and 2.876 (CI 1.367–6.053; *p* < 0.000), respectively. Subjects with SJH were nearly three times more prone to LBPWF and LBPOF. (Table 2).

## 4. Discussion

To the best of our knowledge, this is the largest cohort studying the association between SJH and LBP in young adults. Due to its magnitude, this unique, comprehensive database allows the study of a young and healthy national cohort with minor biases. In this study, the prevalence of self-reported LBP is relatively high (3.7%) but most commonly without objective findings on physical examination and imaging studies (94.8%). Only a minor proportion of the self-reported LBP cohort had an objective finding, allowing further cohort subanalysis regarding SJH association.

In contrast, previous studies [14,15,16] reported various SJH prevalences, ranging from 1% to 25.4% in males and 7% to 38.5% in asymptomatic females [17,18], depending on age, ethnicity, and diagnostic criteria used [19]; our cohort only included subjects who were diagnosed with joint hypermobility by a rheumatologist, explaining the low prevalence that we reported. Due to our cohort size, the formally diagnosed, relatively rare SJH cohort (0.11%) remained large enough to allow investigation into its association with LBP with high confidence levels. In our cohort, SJH was strongly associated with LBP. Subjects with SJH were almost three times more susceptible to having LBP (OR = 2.912, CI 2.437–3.479; *p* < 0.000). The pathophysiological aspects of this association were discussed in several studies. Still, firm evidence of the processes leading to symptoms in subjects with SJH and, more specifically, LBP symptoms are lacking [5,23,24].

Several studies [30,31] proved this association, while others could not prove this association [32]. In this sizeable cross-population study, gender was not associated with symptomatic joint hypermobility (OR = 1.098, CI 0.989–1.223; *p* = 0.09).

LBP was a common condition in this large population national survey of adolescents. While LBP can be associated with significant pathology, such presentation is rare, with symptoms usually mild, nonspecific, and self-limiting. Investigating the association between SJH and LBP, with and without objective findings, we found an association between SJH and both LBP groups.

The study’s limitations include its cross-sectional nature, with the inherent inability to confirm causality or explain the pathophysiology of the findings. Another limitation is the single age group examined (17 years of age). A further limitation is the exclusion of orthodox Jews and other minorities that do not recruit to the IDF. This study did not evaluate data concerning other predisposing factors of LBP, such as BMI. However, it remains valid since cross-sectional studies on such a scale with a structured subject appraisal are complex and rare. Mild or asymptomatic joint hypermobility subjects were not included due to the workup procedure, leading to a possible underestimation of the phenomenon and the lower prevalence in this study compared to other published works.

In conclusion, the results of this sizeable study validate the association between the relatively rare SJH and LBP that has previously been suggested but not established with such significant numbers. In this study, gender was not associated with SJH, but SJH is associated with LBP, with and without objective imaging findings. When evaluating a patient with LBP, one has to assess joint hypermobility as a possible cause of pain and address it in the treatment protocol, limiting stretching and range-of-motion exercises and emphasizing core muscle strengthening and controlled joint motion [14,33]. Further studies are yet to elucidate the exact mechanism leading to the relationship between SJH and LBP that would help understand the impact of specific proprioceptive physiotherapy and stability-oriented treatments as better ways to address patients with back pain related to symptomatic joint hypermobility.

## Figures and Tables

**Table 1 jcm-11-05105-t001:** Symptomatic Joints Hypermobility and Low Back Pain Prevalence in Adolescents.

	Total No.	No LBP		LBP		LBPWF ^a^		LBPOF ^b^	
		No.	%	No.	%	No.	%	No.	%
**No Hypermobility**	1,218,718	1,174,098	96.3%	44,620	3.7%	42,278	3.5%	2342	0.2%
**Hypermobility**	1355	1220	90.0%	135	10.0%	128	9.5%	7	0.5%
**Total**	1,220,073	1,175,318	96.3%	44,755	3.7%	42,406	3.5%	2349	0.2%

^a^ Low back pain without clinical or imaging corroboration. ^b^ Low back pain with clinical or imaging corroboration.

**Table 2 jcm-11-05105-t002:** Odds Ratios for Low Back Pain in Relationship to Symptomatic Joints Hypermobility in Adolescents.

	LBP ^c^			LBPWF ^a^			LBPOF ^b^		
	OR ^c^	95% CI	*p* Value	OR ^d^	95% CI	*p* Value	OR ^d^	95% CI	*p* Value
**Odds Ratio for** **Hypermobility**	**2.912**	**2.437–3.479**	***p* < 0.000**	**2.914**	**2.428–3.497**	***p* < 0.000**	**2.876**	**1.367–6.053**	***p* < 0.000**

Abbreviations: CI, confidence interval; OR, odds ratio. ^a^ Low back pain without clinical or imaging corroboration. ^b^ Low back pain with clinical or imaging corroboration. ^c^ Odds ratios from binary logistic regression. ^d^ Odds ratios from multinomial logistic regression with no low back pain as the base category.

## Data Availability

The Complete data is available under a confidentiality restriction.

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
