# Peer review of "Symptomatic Joint Hypermobility Is Associated with Low Back Pain: A National Adolescents Cohort Study"

_jcm, 2022, doi:10.3390/jcm11175105_

Round 1

Reviewer 1 Report

Manuscript Number: JMC-1866692

Symptomatic Joint Hypermobility is associated with Back pain. A national adolescents cohort study.

Reviewer : 

This is a retrospective epidemiological adolescents cohort study, including 1,220,073 adolescents. This manuscript, entitled "Symptomatic Joint Hypermobility is associated with Back pain. A national adolescents cohort study" aimed to examine the association between symptomatic joint hypermobility (SJH) and LBP. I think the data are interesting. Article appropriate to JMC. Research question was relevant. Cited literature pertinent. Appropriate data analysis. Discussion clear with new insights. Recommendation: Accept with justification of clinical relevance of this manuscript and include the manuscript hypothesis.

Thank you for giving me the opportunity to review this manuscript. This manuscript is quite interesting and present new data. I found the paper interesting to read and congratulate the authors for this effort.

Before publication, some  issues should be addressed:

Introduction:

  • Please, to improve the background and rationale, mention previous studies. "Hypermobility might increase the spinal motion segment's mechanical loads through an increased range of motion”. How? Explain. 
  • "Joint hypermobility is more prevalent in young adults, especially in women". Why gender was not associated with symptomatic joint hypermobility in this study?. Explain. Include in Discussion. 
  • Include the manuscript hypothesis.
  • This paper presented a new idea which seldom been mentioned. Improve the organization of your paper using the following guidelines. The final paragraph of the Introduction should explicitly;
  • The clinical relevance and the rationale for the study need to be strengthened. What's new in the scientific literature with this manuscript? Improve.

Methods:

  • "Data from a medical database containing 17-year-old candidates' records before recruitment into mandatory military service". Please, describe a bit more some characteristics from population. Participant selection.  
  • The study's data was extracted from the Israeli Medical Corps database, as approved by the Israel Defense Forces Medical Corps institutional review board. Ethics Committee approval? What's the record?
  • "Subjects diagnosed with LBP were classified into two severity groups group A had no corroborative objective findings (e.g., neurological deficit or radicular irritation symptoms) on physical examination or imaging studies (i.e., computerized tomography, magnetic resonance imaging, or myelography). In contrast, group B had objective findings correlating to the patient's diagnosis (e.g., herniated disk or spinal stenosis). Why used this classification? Diagnostic assessment? 
  • "A two-step process diagnosed SJH; the first was the medical questionnaire filled by the adolescent stating hypermobility with a documented history of relevant clinical features (joint pain, back pain, neck pain, recurrent ankle sprains, etc.). The second step was hypermobility confirmation by Beighton's score of 5/9 or higher on physical examination by a rheumatologist. It is important to support the use of the instrument and clearly and objectively justify the reason for performing this type of analysis.
  • Discussion 
  • Include the implications/applications of the study for the field movement sciences.    
  •  

Author Response

August 2022

COVER LETTER

I am enclosing a revised manuscript entitled "Symptomatic Joint Hypermobility is associated with Low Back pain, A national adolescents cohort study" submitted to the "Journal of Clinical Medicine" for possible evaluation after extensive revision based on your reviewers' comments remarks for possible evaluation.

With the submission of this manuscript, I would like to undertake that the abovementioned manuscript has not been published elsewhere, accepted for publication elsewhere or under editorial review for publication elsewhere.

Type of Submitted manuscript:

  • Original work

I want to share the following information with Editor-in-Chief:

"Low Back pain (LBP) is one of the most common medical complaints affecting many worldwide and costing billions. Studies suggest a link between LBP and joint hypermobility. This study aimed to examine the association between symptomatic joint hypermobility (SJH) and LBP.

Our unique database allowed us to calculate the prevalence of SJH in 1,220,073 young adults and identified 1355 cases (0.111%). LBP complaints were identified in 44755 subjects (3.7%). When examining SJH and LBP's association, the Odds Ratios were 2.912 (CI 2.437-3.479, P<0.000). Thus, a strong association between SJH and LBP was identified. Furthermore, subjects with SJH were nearly three times more subjected to LBP mild and severe.

According to our study, SJH is strongly associated with LBP symptoms in young adults.

The results of this sizeable study validate the association between SJH and LBP that has been previously suggested but not established with such significant numbers. When evaluating a patient with LBP, one has to assess joint hypermobility as a possible cause of pain and address it in the treatment protocol, limiting stretching and range of motion exercises and emphasizing core muscle strengthening and controlled joint motion. The exact mechanism leading to the relationship between SJH and LBP is yet to be elucidated by further studies that would help understand the impact of specific proprioceptive physiotherapy and stability-oriented treatments as better ways to address patients with back pain related to symptomatic joint hypermobility."

We believe that the important findings of this work can contribute to a better understanding of the association between SJH and LBP with a significant clinical impact and be an excellent fit for the Section: Epidemiology & Public Health; Special Issue - Physiotherapy in Muscle Pain: Current Updates from Theory to Clinical Practice.

As such, we hope we can publish this work in the journal.

On behalf of all the authors,

Sincerely yours

The Corresponding Author

Symptomatic Joint Hypermobility is associated with Back pain. A national adolescents cohort study.

Dear Reviewers,

We appreciate your efforts in making our paper better and suitable for publication in the Journal of Clinical Medicine.

Please find our point-to-point replies to your queries.

All changes in the text are underlined and bolded. 

Reviewer 1:

This is a retrospective epidemiological adolescents cohort study, including 1,220,073 adolescents. This manuscript, entitled "Symptomatic Joint Hypermobility is associated with Back pain. A national adolescents cohort study" aimed to examine the association between symptomatic joint hypermobility (SJH) and LBP. I think the data are interesting. Article appropriate to JMC. Research question was relevant. Cited literature pertinent. Appropriate data analysis. Discussion clear with new insights. Recommendation: Accept with justification of clinical relevance of this manuscript and include the manuscript hypothesis.

Thank you for giving me the opportunity to review this manuscript. This manuscript is quite interesting and present new data. I found the paper interesting to read and congratulate the authors for this effort.

Before publication, some  issues should be addressed:

Thank you for your efforts and input. We have answered all your points as explained below.

Introduction:

  • Please, to improve the background and rationale, mention previous studies. "Hypermobility might increase the spinal motion segment's mechanical loads through an increased range of motion". How? Explain. 

Please see lines 46-47 marked; we have added information regarding the mechanism of increased spinal motion mechanical loads.

  • "Joint hypermobility is more prevalent in young adults, especially in women". Why gender was not associated with symptomatic joint hypermobility in this study?. Explain. Include in Discussion. 

Some papers found gender associated with SJH, while others did not find it associated with it. We have included both and added a paragraph in the discussion section (underlined) – Lines 171-173.

  • Include the manuscript hypothesis.

We added a Hypothesis at the end of the introduction section (line 75).

  • This paper presented a new idea which seldom been mentioned. Improve the organization of your paper using the following guidelines. The final paragraph of the Introduction should explicitly;

We have reorganized the paper accordingly.

  • The clinical relevance and the rationale for the study need to be strengthened. What's new in the scientific literature with this manuscript? Improve.

The discussion was strengthened and elaborated further.

All changes in the text are underlined and bolded.  

Methods:

  • "Data from a medical database containing 17-year-old candidates' records before recruitment into mandatory military service". Please, describe a bit more some characteristics from population. Participant selection.  

As a large database study, we included all characteristics available.

Participants selection is described in detail in lines 78-87.

  • The study's data was extracted from the Israeli Medical Corps database, as approved by the Israel Defense Forces Medical Corps institutional review board. Ethics Committee approval? What's the record?

IRB form was added to the files.

IRB APPROVAL No. 947-2010-1 given by the IDF IRB.

  • "Subjects diagnosed with LBP were classified into two severity groups group A had no corroborative objective findings (e.g., neurological deficit or radicular irritation symptoms) on physical examination or imaging studies (i.e., computerized tomography, magnetic resonance imaging, or myelography). In contrast, group B had objective findings correlating to the patient's diagnosis (e.g., herniated disk or spinal stenosis). Why used this classification? Diagnostic assessment? 

While LBP is ubiquitous, only a few cases have objective findings requiring intervention. Understanding LBP complaints versus imaging findings will help with understanding the true impact of SJH on LBP patients.

  • "A two-step process diagnosed SJH; the first was the medical questionnaire filled by the adolescent stating hypermobility with a documented history of relevant clinical features (joint pain, back pain, neck pain, recurrent ankle sprains, etc.). The second step was hypermobility confirmation by Beighton's score of 5/9 or higher on physical examination by a rheumatologist. It is important to support the use of the instrument and clearly and objectively justify the reason for performing this type of analysis.

We used a medical questionnaire filled by the adolescents as a screening tool, followed by Beighton's score of 5/9 or higher on physical examination by a rheumatologist to confirm the diagnosis of SJH.

We have enforced that point in the methods section.   

  • Discussion 
  • Include the implications/applications of the study for the field movement sciences.

We have included in the conclusion sentence regarding possible clinical implications of this study (lines 194-198).     

Thank you for your efforts and input.

We have answered all your points as explained above.

Thank you for your kind words.

We hope this revised version can now be published in JCM.

On behalf of all the authors,

Sincerely yours

The Corresponding Author

Reviewer 2 Report

It is an interesting topic, but there are some major concerns.

Overall: The use of mild and severe is not in accordance with the tables it says there less severe and more severe? This is also the case in more parts of the manuscript and it is important to be consequent in the text and in the tables. Perhaps is the use of mild and severe better then less and more severe? I have bold mild and severe and also less severe and more severe in the comments below. Depending on which terms you use it must be change throughout the manuscript.

I have also a suggestion to define the four groups more clear, see below. 

The title must be more clear: Change Back Pain to Low Back Pain

Abstract

Line 21-24: ..........into mild (3.5%) and severe symptoms (0.2%). The association between SJH and LBP was examined: Odds Ratio (OR) was 2.912 ( P<0.0001). The OR was further calculated for mild and severe LBP, 2.914 ( P<0.000) and 2.876 (P<0.000), respectively. Subjects with SJH were almost three times more prone to mild and severe LBP.

Introduction

The content of introduction is OK, but I question the use of the references, for example references 1 is about "Magnetic resonance imaging....." and  2 about "Associations of body mass index......." but they are used as references for the sentence  "The prevalence of LBP in all age groups is still controversial but 38 relatively high [1, 2]". As I can see the two articles have information in relation to prevalence in their introduction, but then the references in their introduction must be used.

Another example is references 26 and 27. They are in relation to "Intrarater and interrater reliability of the Beighton.....(26) and "Beighton Score: A Valid Measure for Generalized Hypermobility in Children", but used in this manuscript for the sentence "Association between LBP and SJH is still an open 69 debate that justifies further studies into the possible pathophysiological basis [26, 27]". 

It is necessary to go through all references in the introduction.

Material and Methods

The evaluation process includes a lot of different questionnaires and reports. It is necessary to develop this part. There is a need to know more in details the content of the different questionnaires and reports. Now it is to fragmentary. 

There is also a need to clearly define the four groups "No Low Back Pain", "Low Back Pain", "Low Back Pain (Less Severe)" and "Low Back Pain (More Severe)". Now it is difficult to know the characteristics of the groups, what are the differences. 

Line 92; what does 2 in brackets stands for?

Line 100; Beighton's score there is a need to add the origin reference, "Beighton P, Solomon L, Soskolne CL. Articular mobility in an African population. Ann Rheum Dis. 1973;32:413–8"

Line 101-102; Add references to Marfan's disease ( ) or Ehlers-Danlos syndrome ( ).

Table 1.

Add % in the first column in relation to total number. In the column "Number of no Low back pain" write 11740 98 and 11753 18 in one line (now in two lines). The same with the last column in relation to "Low Back Pain (More Severe)" 23 42 and 23 49 in one line.

Line 114;  3.7%..... Rewrite the sentence, so that it starts with letters.

Line 117; .....severe LBP, with correlating anatomical findings. Difficult to understand "with correlating anatomical findings", rewrite or delete.

Line 119; Most 1,220 subjects had mild LBP without objective findings. This sentence is unclear, because of that there is no definitions in relation to the four groups, see above.

Line 119-120;  Severe LBP was rare in SJH and non-SJH groups of adolescents (0.2%). Add 0.5% in the brackets (0.2% respectively 0.5%).

Line 120; ......but 2.5 more prevalent in the SJH group of patients (p < 0.0001) (Table 1). I can't see this result in Table 1?. Reference to Table 1 should be moved to after .....non-SJH groups of adolescents (0.2%) (Table 1) and then do a line break and start with a sentence in relation to that 2.5 more prevalent in the ..........

Line 130; ......"according to LBP severity, mild or severe, resulted..." Should be change to less severe and more severe, as it says in the Table 2. Or use mild and severe, depending on comment above.

Line 132; ...."three times more prone to mild and severe LBP". Should be change to less severe and more severe, as it says in the Table 2. Or use mild and severe, depending on comment above.

Discussion

Also in the discussion there is a need to go through the used references. For example line 145, "As previously reported [2],..." the reference 2 is in relation to BMI?

References

Reference 24 should be written in low cases.

The references must be written in accordance to the instructions of the journal, for example if there is "a large number of persons (more than 10 authors), you can either cite all authors, or cite the first ten authors, then add a semicolon and add ‘et al.’ at the end" Now many references is written for example reference 1 Jensen, M.C., et al., .... 2. Hershkovich, O., et al.,

Author Response

August 2022

COVER LETTER

I am enclosing a revised manuscript entitled "Symptomatic Joint Hypermobility is associated with Low Back pain, A national adolescents cohort study" submitted to the "Journal of Clinical Medicine" for possible evaluation after extensive revision based on your reviewers' comments remarks for possible evaluation.

With the submission of this manuscript, I would like to undertake that the abovementioned manuscript has not been published elsewhere, accepted for publication elsewhere or under editorial review for publication elsewhere.

Type of Submitted manuscript:

  • Original work

I want to share the following information with Editor-in-Chief:

"Low Back pain (LBP) is one of the most common medical complaints affecting many worldwide and costing billions. Studies suggest a link between LBP and joint hypermobility. This study aimed to examine the association between symptomatic joint hypermobility (SJH) and LBP.

Our unique database allowed us to calculate the prevalence of SJH in 1,220,073 young adults and identified 1355 cases (0.111%). LBP complaints were identified in 44755 subjects (3.7%). When examining SJH and LBP's association, the Odds Ratios were 2.912 (CI 2.437-3.479, P<0.000). Thus, a strong association between SJH and LBP was identified. Furthermore, subjects with SJH were nearly three times more subjected to LBP mild and severe.

According to our study, SJH is strongly associated with LBP symptoms in young adults.

The results of this sizeable study validate the association between SJH and LBP that has been previously suggested but not established with such significant numbers. When evaluating a patient with LBP, one has to assess joint hypermobility as a possible cause of pain and address it in the treatment protocol, limiting stretching and range of motion exercises and emphasizing core muscle strengthening and controlled joint motion. The exact mechanism leading to the relationship between SJH and LBP is yet to be elucidated by further studies that would help understand the impact of specific proprioceptive physiotherapy and stability-oriented treatments as better ways to address patients with back pain related to symptomatic joint hypermobility."

We believe that the important findings of this work can contribute to a better understanding of the association between SJH and LBP with a significant clinical impact and be an excellent fit for the Section: Epidemiology & Public Health; Special Issue - Physiotherapy in Muscle Pain: Current Updates from Theory to Clinical Practice.

As such, we hope we can publish this work in the journal.

On behalf of all the authors,

Sincerely yours

The Corresponding Author

Symptomatic Joint Hypermobility is associated with Back pain. A national adolescents cohort study.

Dear Reviewers,

We appreciate your efforts in making our paper better and suitable for publication in the Journal of Clinical Medicine.

Please find our point-to-point replies to your queries.

All changes in the text are underlined and bolded. 

Reviewer 2:

It is an interesting topic, but there are some major concerns.

Overall: The use of mild and severe is not in accordance with the tables it says there less severe and more severe? This is also the case in more parts of the manuscript and it is important to be consequent in the text and in the tables. Perhaps is the use of mild and severe better then less and more severe? I have bold mild and severe and also less severe and more severe in the comments below. Depending on which terms you use it must be change throughout the manuscript.

I have also a suggestion to define the four groups more clear, see below. 

We agree on this entirely.

Based on your suggestions, we have changed the terms in the paper to – LBP without objective findings (LBPWF) and LBP with objective findings (LBPOF).

All changes are underlined in the text.

The title must be more clear: Change Back Pain to Low Back Pain

Corrected.

Abstract

Line 21-24: ..........into mild (3.5%) and severe symptoms (0.2%). The association between SJH and LBP was examined: Odds Ratio (OR) was 2.912 ( P<0.0001). The OR was further calculated for mild and severe LBP, 2.914 ( P<0.000) and 2.876 (P<0.000), respectively. Subjects with SJH were almost three times more prone to mild and severe LBP.

Based on your suggestions, we have changed the terms in the paper to – LBP without objective findings (LBPWF) and LBP with objective findings (LBPOF).

All changes are underlined in the text.

Introduction

The content of introduction is OK, but I question the use of the references, for example references 1 is about "Magnetic resonance imaging....." and  2 about "Associations of body mass index......." but they are used as references for the sentence  "The prevalence of LBP in all age groups is still controversial but 38 relatively high [1, 2]". As I can see the two articles have information in relation to prevalence in their introduction, but then the references in their introduction must be used.

Another example is references 26 and 27. They are in relation to "Intrarater and interrater reliability of the Beighton.....(26) and "Beighton Score: A Valid Measure for Generalized Hypermobility in Children", but used in this manuscript for the sentence "Association between LBP and SJH is still an open 69 debate that justifies further studies into the possible pathophysiological basis [26, 27]". 

It is necessary to go through all references in the introduction.

All references were examined and corrected if indicated. The list was updated and corrected accordingly.

Material and Methods

The evaluation process includes a lot of different questionnaires and reports. It is necessary to develop this part. There is a need to know more in details the content of the different questionnaires and reports. Now it is to fragmentary. 

Recruits fill out a standard general health questionnaire that includes a section regarding hypermobility symptoms. Following this form, if indicated, a further medical assessment is added. This is described in the methods section, similar to previous publications on this database.

There is also a need to clearly define the four groups "No Low Back Pain", "Low Back Pain", "Low Back Pain (Less Severe)" and "Low Back Pain (More Severe)". Now it is difficult to know the characteristics of the groups, what are the differences. 

We agree.

Based on your suggestions, we have changed the terms in the paper to – LBP without objective findings (LBPWF) and LBP with objective findings (LBPOF).

All changes are underlined in the text.

Line 92; what does 2 in brackets stands for?

This is a reference – not required here and has been removed.

Line 100; Beighton's score there is a need to add the origin reference, "Beighton P, Solomon L, Soskolne CL. Articular mobility in an African population. Ann Rheum Dis. 1973;32:413–8"

Line 101-102; Add references to Marfan's disease ( ) or Ehlers-Danlos syndrome ( ).

Added - changes are underlined in the text. 

Table 1.

Add % in the first column in relation to total number. In the column "Number of no Low back pain" write 11740 98 and 11753 18 in one line (now in two lines). The same with the last column in relation to "Low Back Pain (More Severe)" 23 42 and 23 49 in one line.

Line 114;  3.7%..... Rewrite the sentence, so that it starts with letters.

Line 117; .....severe LBP, with correlating anatomical findings. Difficult to understand "with correlating anatomical findings", rewrite or delete.

Line 119; Most 1,220 subjects had mild LBP without objective findings. This sentence is unclear, because of that there is no definitions in relation to the four groups, see above.

Line 119-120;  Severe LBP was rare in SJH and non-SJH groups of adolescents (0.2%). Add 0.5% in the brackets (0.2% respectively 0.5%).

Line 120; ......but 2.5 more prevalent in the SJH group of patients (p < 0.0001) (Table 1). I can't see this result in Table 1?. Reference to Table 1 should be moved to after .....non-SJH groups of adolescents (0.2%) (Table 1) and then do a line break and start with a sentence in relation to that 2.5 more prevalent in the ..........

Line 130; ......"according to LBP severity, mild or severe, resulted..." Should be change to less severe and more severe, as it says in the Table 2. Or use mild and severe, depending on comment above.

Line 132; ...."three times more prone to mild and severe LBP". Should be change to less severe and more severe, as it says in the Table 2. Or use mild and severe, depending on comment above.

Based on your comments, we have changed the text for each point.

All changes are underlined in the text.

Discussion

Also in the discussion there is a need to go through the used references. For example line 145, "As previously reported [2],..." the reference 2 is in relation to BMI?

Reference 2 refers to the incidence of LBP in the adolescent population and is relevant to the subject discussed.

All references were examined and corrected if indicated. The list was updated and corrected accordingly.

References

Reference 24 should be written in low cases.

Corrected.

The references must be written in accordance to the instructions of the journal, for example if there is "a large number of persons (more than 10 authors), you can either cite all authors, or cite the first ten authors, then add a semicolon and add 'et al.' at the end" Now many references is written for example reference 1 Jensen, M.C., et al., .... 2. Hershkovich, O., et al.,

Corrected.

Thank you for your efforts and input.

We have answered all your points as explained above.

Thank you for your kind words.

We hope this revised version can now be published in JCM.

On behalf of all the authors,

Sincerely yours

The Corresponding Author

Reviewer 3 Report

Review: Symptomatic Joint Hypermobility is associated with Back pain, A national adolescents cohort study 3 

Association between hypermobility and back pain is often seen in the clinical practice, but I have not seen studies on how much of a risk factor for low back pain hypermobility is. This study tries to explore this. The problem is that the prevalence of low back pain is around 7.5% globally while the prevalence in this study was only 3.7% and the prevalence of low back pain among those with symptomatic joint hypermobility in the study was 11%. This makes the calculations very theoretical, especially when the results show an odds ratio of not more than 2.9 (and a relative risk of 2.72)

The authors discuss the fact that a majority of those with low back pain had no radiological findings or objective findings when examined – which is the normal situation since pain is a subjective experience that is not possible to visualize. The division in subgroups might still be meaningful but it is wrong to call the groups more or less severe based on radiological findings!

Register studies can be interesting despite their lack of deeper understanding on the pathophysiology and correlation mechanisms between risk factors and disease, but I would like to see a discussion on the selection to the database and the exclusion criteria. They mention that Marfans syndrome and EDS was excluded. I would also like to see data on the known risk factors for low back pain: BMI.

The conclusion that hypermobility is associated with low back pain I feel is going too far based on these data. ”Higher risk for low back pain if having symptomatic joint hypermobility” is the furthest that I would say, and with a prevalence as low as 0.11% of hypermobility in the population studied, this becomes a theoretical finding more than a clinically relevant finding!

Finally, they mention that the study was ”approved by Israel defense Forces Medical Corps institutional review board”. Was this a formal ethical approval or was it only an approval that they had permission to study the register? If it was an ethical approval, there should be a diary number, please. And a post in any of the official registers like clinicaltrials.gov would have been appreciated!

But the article is well written and the language nice. Might be worth publishing after a major revision if the questions about ethical approval are solved and more appropriate conclusions are drawn.

Author Response

August 2022

COVER LETTER

I am enclosing a revised manuscript entitled "Symptomatic Joint Hypermobility is associated with Low Back pain, A national adolescents cohort study" submitted to the "Journal of Clinical Medicine" for possible evaluation after extensive revision based on your reviewers' comments remarks for possible evaluation.

With the submission of this manuscript, I would like to undertake that the abovementioned manuscript has not been published elsewhere, accepted for publication elsewhere or under editorial review for publication elsewhere.

Type of Submitted manuscript:

  • Original work

I want to share the following information with Editor-in-Chief:

"Low Back pain (LBP) is one of the most common medical complaints affecting many worldwide and costing billions. Studies suggest a link between LBP and joint hypermobility. This study aimed to examine the association between symptomatic joint hypermobility (SJH) and LBP.

Our unique database allowed us to calculate the prevalence of SJH in 1,220,073 young adults and identified 1355 cases (0.111%). LBP complaints were identified in 44755 subjects (3.7%). When examining SJH and LBP's association, the Odds Ratios were 2.912 (CI 2.437-3.479, P<0.000). Thus, a strong association between SJH and LBP was identified. Furthermore, subjects with SJH were nearly three times more subjected to LBP mild and severe.

According to our study, SJH is strongly associated with LBP symptoms in young adults.

The results of this sizeable study validate the association between SJH and LBP that has been previously suggested but not established with such significant numbers. When evaluating a patient with LBP, one has to assess joint hypermobility as a possible cause of pain and address it in the treatment protocol, limiting stretching and range of motion exercises and emphasizing core muscle strengthening and controlled joint motion. The exact mechanism leading to the relationship between SJH and LBP is yet to be elucidated by further studies that would help understand the impact of specific proprioceptive physiotherapy and stability-oriented treatments as better ways to address patients with back pain related to symptomatic joint hypermobility."

We believe that the important findings of this work can contribute to a better understanding of the association between SJH and LBP with a significant clinical impact and be an excellent fit for the Section: Epidemiology & Public Health; Special Issue - Physiotherapy in Muscle Pain: Current Updates from Theory to Clinical Practice.

As such, we hope we can publish this work in the journal.

On behalf of all the authors,

Sincerely yours

The Corresponding Author

Symptomatic Joint Hypermobility is associated with Back pain. A national adolescents cohort study.

Dear Reviewers,

We appreciate your efforts in making our paper better and suitable for publication in the Journal of Clinical Medicine.

Please find our point-to-point replies to your queries.

All changes in the text are underlined and bolded. 

Reviewer 3:

Review: Symptomatic Joint Hypermobility is associated with Back pain, A national adolescents cohort study 3 

Association between hypermobility and back pain is often seen in the clinical practice, but I have not seen studies on how much of a risk factor for low back pain hypermobility is. This study tries to explore this. The problem is that the prevalence of low back pain is around 7.5% globally while the prevalence in this study was only 3.7% and the prevalence of low back pain among those with symptomatic joint hypermobility in the study was 11%. This makes the calculations very theoretical, especially when the results show an odds ratio of not more than 2.9 (and a relative risk of 2.72)

 The authors discuss the fact that a majority of those with low back pain had no radiological findings or objective findings when examined – which is the normal situation since pain is a subjective experience that is not possible to visualize. The division in subgroups might still be meaningful but it is wrong to call the groups more or less severe based on radiological findings!

We agree on this entirely.

Based on your suggestions, we have changed the terms in the paper to – LBP without objective findings (LBPWF) and LBP with objective findings (LBPOF). All changes are underlined in the text. 

Register studies can be interesting despite their lack of deeper understanding on the pathophysiology and correlation mechanisms between risk factors and disease, but I would like to see a discussion on the selection to the database and the exclusion criteria.

They mention that Marfans syndrome and EDS was excluded. I would also like to see data on the known risk factors for low back pain: BMI.

We added further limitations to this study regarding the above points.

All changes are underlined in the text. 

The conclusion that hypermobility is associated with low back pain I feel is going too far based on these data." Higher risk for low back pain if having symptomatic joint hypermobility" is the furthest that I would say, and with a prevalence as low as 0.11% of hypermobility in the population studied, this becomes a theoretical finding more than a clinically relevant finding!

Mild or asymptomatic joint hypermobility subjects were not included due to the workup procedure, leading to a possible underestimation of the phenomenon and the lower prevalence in this study compared to other published work. 

"When evaluating a patient with LBP, one has to assess joint hypermobility as a possible cause of pain and address it in the treatment protocol…" - Lines 195-198.

This is the clinical relevance.

Changes are underlined in the text. 

Finally, they mention that the study was" approved by Israel defense Forces Medical Corps institutional review board". Was this a formal ethical approval or was it only an approval that they had permission to study the register? If it was an ethical approval, there should be a diary number, please. And a post in any of the official registers like clinicaltrials.gov would have been appreciated!

IRB form was added to the files.

IRB APPROVAL No. 947-2010-1. 

But the article is well written and the language nice. Might be worth publishing after a major revision if the questions about ethical approval are solved and more appropriate conclusions are drawn.

Thank you for your efforts and input.

We have answered all your points as explained above.

Thank you for your kind words.

We hope this revised version can now be published in JCM.

On behalf of all the authors,

Sincerely yours

The Corresponding Author

Round 2

Reviewer 2 Report

The paper is much improved, but I have still some comments:

Line  40 and 41: The prevalence of LBP in all age groups is still controversial but relatively high [1,2]. You referer to study 1 and 2. The study 1 is in relation to BMI and height. Reference 2 is good. The reference 1 can be taken away.

Line 55: In reference 17 I can´t find anything in relation to 0.6%.

Line 55: Difficult to understand the figure 31.5% from reference 18.

Line 119-121: Three point seven percent of the cohort (44,755 adolescents) suffered from LBP, and 0.11% (1,355 120 adolescents) suffered SJH. Perhaps rewrite the sentence for not starting with Three point seven percent.

Line 154-157:  As previously reported [1], the prevalence of self-reported LBP is relatively high (3.7%) but most commonly (94.6%) without objective findings on physical examination and imaging studies. It seems that the figures are from this study and not from reference 1?

Line 161: .....from 0.6% [17] to 31.5% [18].. See above in relation to the figures?

Abstract: In relation to comments from one of the reviewer the aim has change also including gender differences. This must also be written in the aim and results in the abstract. 

Methods:

I still think a bit more information is needed in relation to above all the questionnarie. 

Reference 24: LARSSON, L.G.; Mudholkar, G.S.; Baum, J.; Srivastava, D.J.J.o.i.m. Benefits and liabilities of hypermobility in the back pain 250 disorders of industrial workers. 1995, 238, 461-467. Larsson must be written in small letters

The whole reference list is confusing in the way it is written. Just some examples:

Reference  3 . Freburger, J.K.; Holmes, G.M.; Agans, R.P.; Jackman, A.M.; Darter, J.D.; Wallace, A.S.; Castel, L.D.; Kalsbeek, W.D.; Carey, 211 T.S. The rising prevalence of chronic low back pain. Archives of internal medicine 2009, 169, 251-258, 212 doi:10.1001/archinternmed.2008.543. This reference has the journal printed out in full text, not abbreviated?

Reference 15.Wolf, J.M.; Cameron, K.L.; Owens, B.D.J.J.-J.o.t.A.A.o.O.S. Impact of joint laxity and hypermobility on the musculoskeletal 234 system. 2011, 19, 463-471 There is the journal abbreviated after the authors? with J.o.t.A.A.o.O.S?

Important to follow the journals reference system.   

Author Response

Dear Reviewers,

We appreciate your efforts in making our paper better and suitable for publication in the Journal of Clinical Medicine.

Please find our point-to-point replies to your queries.

All changes in the text are underlined and bolded.  

Reviewer 2: Round 2:

The paper is much improved, but I have still some comments:

Line  40 and 41: The prevalence of LBP in all age groups is still controversial but relatively high [1,2]. You referer to study 1 and 2. The study 1 is in relation to BMI and height. Reference 2 is good. The reference 1 can be taken away.

Reference 1 is very relevant to the incidence of LBP in adolescents since it screened its prevalence in 830,000 adolescents.

Line 55: In reference 17 I can´t find anything in relation to 0.6%.

Line 55: Difficult to understand the figure 31.5% from reference 18.

We have precisely rewritten the percentage, as mentioned in the references, separating females and males.

Line 119-121: Three point seven percent of the cohort (44,755 adolescents) suffered from LBP, and 0.11% (1,355 120 adolescents) suffered SJH. Perhaps rewrite the sentence for not starting with Three point seven percent.

We have rewritten the sentence accordingly.

Line 154-157:  As previously reported [1], the prevalence of self-reported LBP is relatively high (3.7%) but most commonly (94.6%) without objective findings on physical examination and imaging studies. It seems that the figures are from this study and not from reference 1?

We agree; reference 1 was removed, and the sentence was rewritten accordingly, referring to the results of this study (Table 1). 

Line 161: .....from 0.6% [17] to 31.5% [18].. See above in relation to the figures?

We have precisely rewritten the percentage, as mentioned in the references, separating females and males.

Abstract: In relation to comments from one of the reviewer the aim has change also including gender differences. This must also be written in the aim and results in the abstract. 

We have rewritten the sentence accordingly.

Methods:

I still think a bit more information is needed in relation to above all the questionnarie. 

We added more information about this questionnaire; please see - Lines 84-84.

Reference 24: LARSSON, L.G.; Mudholkar, G.S.; Baum, J.; Srivastava, D.J.J.o.i.m. Benefits and liabilities of hypermobility in the back pain 250 disorders of industrial workers. 1995, 238, 461-467. Larsson must be written in small letters

The whole reference list is confusing in the way it is written. Just some examples:

Reference  3 . Freburger, J.K.; Holmes, G.M.; Agans, R.P.; Jackman, A.M.; Darter, J.D.; Wallace, A.S.; Castel, L.D.; Kalsbeek, W.D.; Carey, 211 T.S. The rising prevalence of chronic low back pain. Archives of internal medicine 2009, 169, 251-258, 212 doi:10.1001/archinternmed.2008.543. This reference has the journal printed out in full text, not abbreviated?

Reference 15.Wolf, J.M.; Cameron, K.L.; Owens, B.D.J.J.-J.o.t.A.A.o.O.S. Impact of joint laxity and hypermobility on the musculoskeletal 234 system. 2011, 19, 463-471 There is the journal abbreviated after the authors? with J.o.t.A.A.o.O.S?

Important to follow the journals reference system.   

References were corrected accordingly.

Reviewer 3 Report

After the revision the manuscript is very much improved. The conclusions are now more appropriate. I still wish the correlation to BMI would have been included, but the lack of association with gender is interesting but a table showing BMI, gender and prevalence of SJH and LBP would help giving a better understanding of the population. I suppose there are data on BMI in the database, so it should not be too difficult to include. 

I still wish that the authors in the discussion comment on the fact that the population with SJH is very small. Emphasizing that the study is "sizeable" is not needed as it impies that the problem should be larger that it is, while the truth is that the condition is so rare that you need a large study in order to get any significant results!

I am relieved that there was an ethical approval, but that should be better described in the methods. There should be a paragraph on ethical approval where diary number is given and the authors should mention why there were no informed consent from the subjects and how the subjects integrity was secured without informed consent. If including the approval from the Israel defense forces medical corps in supplement, an English translation should be added.

Author Response

Reviewer 3: Round 2:

After the revision the manuscript is very much improved. The conclusions are now more appropriate. I still wish the correlation to BMI would have been included, but the lack of association with gender is interesting but a table showing BMI, gender and prevalence of SJH and LBP would help giving a better understanding of the population. I suppose there are data on BMI in the database, so it should not be too difficult to include.

Unfortunately, we are currently unable to add information regarding BMI.

I still wish that the authors in the discussion comment on the fact that the population with SJH is very small. Emphasizing that the study is "sizeable" is not needed as it impies that the problem should be larger that it is, while the truth is that the condition is so rare that you need a large study in order to get any significant results!

We have rewritten the sentence accordingly.

Please see the bolded and underlined lines 166-167 and 196.  

I am relieved that there was an ethical approval, but that should be better described in the methods. There should be a paragraph on ethical approval where diary number is given and the authors should mention why there were no informed consent from the subjects and how the subjects integrity was secured without informed consent. If including the approval from the Israel defense forces medical corps in supplement, an English translation should be added.

As a retrospective large data study, no identified medical information was retrieved.

No informed consent is required in such studies.

IDF IRB approved this study with no consent required.

The methods section was rewritten accordingly, Lines 93-95.

A translated IDF IRB form will be available as requested by the editor.   

Thank you for your efforts and input.

We have answered all your points as explained above.

Thank you for your kind words.

We hope this revised version can now be published in JCM.

BW

The corresponding author
